# Site-Directed Spin Labeling EPR Spectroscopy for Determination of Albumin Structural Damage and Hypoalbuminemia in Critical COVID-19

**DOI:** 10.3390/antiox11122311

**Published:** 2022-11-22

**Authors:** Ekaterina Georgieva, Yanka Karamalakova, Georgi Arabadzhiev, Vasil Atanasov, Rositsa Kostandieva, Mitko Mitev, Vanya Tsoneva, Yovcho Yovchev, Galina Nikolova

**Affiliations:** 1Department of “General and Clinical Pathology, Forensic Medicine, Deontology and Dermatovenerology”, Medical Faculty, Trakia University, 11 Armeiska Str., 6000 Stara Zagora, Bulgaria; 2Department of “Medical Chemistry and Biochemistry”, Medical Faculty, Trakia University, 11 Armeiska Str., 6000 Stara Zagora, Bulgaria; 3Department of “Surgery and Anesthesiology”, University Hospital “Prof. Dr. St. Kirkovich”, 6000 Stara Zagora, Bulgaria; 4Forensic Toxicology Laboratory, Military Medical Academy, 3 “Sv. Georgi Sofiiski Str.”, 1606 Sofia, Bulgaria; 5Department of “Diagnostic Imaging”, University Hospital “Prof. Dr. St. Kirkovich”, 6000 Stara Zagora, Bulgaria; 6Department of Propaedeutics of Internal Medicine and Clinical Laboratory, Medical Faculty, Trakia University, 11 Armeiska Str., 6000 Stara Zagora, Bulgaria

**Keywords:** COVID-19, hypoalbuminemia, intoxication, 3-Maleimido-PROXYL, site-directed spin labeling EPR, spin labeling

## Abstract

The main factors in the COVID-19 pathology, which can initiate extensive structural changes at the cellular and molecular levels, are the generation of free radicals in abnormal amounts, and oxidative stress. Under “oxidative shock” conditions, the proteins undergo various modifications that affect their function and activity, and as a result distribute malfunctioning protein derivatives in the body. Human serum albumin is a small globular protein characterized by a high overall binding capacity for neutral lipophilic and acidic dosage forms. The albumin concentration is crucial for the maintenance of plasma oncotic pressure, the transport of nutrients, amino acids, and drugs, the effectiveness of drug therapy, and the prevention of drug toxicity. Hypoalbuminemia and structural defects molecule in the protein suggest a risk of changed metabolism and increased plasma concentration of unbound drugs. Therefore, the albumin structural and functional changes accompanied by low protein levels can be a serious prerequisite for ineffective therapy, frequent complications, and high mortality in patients with SARS-CoV-2 infection. The current opinion aims the research community the application of Site-Directed Spin Labeling Electron Paramagnetic Resonance spectroscopy (SDSL-EPR) and 3-Maleimido-PROXYL radical in determining abnormalities of the albumin dynamics and protein concentrations in COVID-19 critical patients.

## 1. Introduction

An uncontrolled systemic inflammatory response known as cytokine release syndrome (immune and non-immune effector cells) or cytokine storm occurs during coronavirus infection (COVID-19). The released pro-inflammatory cytokines (IL-1β, IL-6, IL-8, IL-17, and TNFα) lead to an extensive inflammatory response that plays a major role in the development of acute damage in the body. Several risk factors such as gender, advanced age, ethnicity, and chronic diseases such as hyperglycemia and obesity are associated with elevated oxidant levels and contribute to severe infection and high mortality [1]. Reactive oxygen species (ROS) are produced as a natural response from normal oxygen metabolism in the body, participate in many cell signaling pathways, and are necessary for eradicating viruses phagocytosed by immune cells. Various viral infections such as respiratory syncytial viruses, rhinoviruses, and various coronaviruses are characterized by the production of abnormal amounts of oxygen radicals [2]. For example, SARS-CoV-2 viral pneumonia causes overactivation of the immune response in lung tissues, which is almost always accompanied by mitochondrial oxidative stress (OS) and subsequent endothelial damage [1]. The adequate response of the body to a viral infection must include a strictly maintained redox homeostasis, as the transition to excessive production of oxidants leads to the development of OS, impaired redox homeostasis, and multiorgan damage. Excess ROS and high levels of oxidative (OS) and nitrosative (NS) stress are known to cause irreversible oxidative damage to important biomacromolecules, membrane phospholipids, and proteins, disrupting their structure and functions [3]. Excess ROS and high levels of oxidative (OS) and nitrosative (NS) stress are known to cause irreversible oxidative damage to important biomacromolecules, membrane phospholipids, and proteins, disrupting their structure and functions [3]. However, ROS are involved in cell signaling in the regulation of cellular processes, thanks to their ability to mediate the reversible oxidation of cysteine. For example, H_2_O_2_ has emerged as the major redox-signaling metabolite capable of mediating the reversible oxidation of thiol groups in proteins. Oxidative modifications of cysteine modulate protein function and are involved in the regulation of intracellular processes in response to external stimuli [4]. In systemic inflammatory processes, there are deviations from the reference values of various biochemical indicators such as white blood cells, neutrophils, lymphocytes, platelets, and D-dimer, which is complemented by another important prognostic marker—serum albumin (HAS). The HAS functionality is affected by various factors, such as changes in the extracellular redox balance in favor of oxidants, which can lead to structural disorders in its molecule, pH, transport of transition metal ions, nitric oxide, hemin, and drugs [5]. Albumin contains 34 cysteine residues, 17 disulfide bridges, and one redox-active free cysteine residue Cys^34^ (reactive nucleophile), which is a major target of oxidative damage [6]. In the acute phase of COVID-19, HAS acts as an antioxidant, but high levels of OS can lead to its irreversible oxidation [6,7]. Free radicals (ROS, RNS) are responsible for oxidative damage to HAS, oxidized albumin and low protein level have been implicated in the severity and mortality of patients with SARS-CoV-2 infection [8]. Badawy et al., reported that hydrogen peroxide and neutrophil-mediated OS induced protein structural and functional changes in patients with COVID-19 [9].

Abnormal HAS levels can impede the active transport of medicaments and increase drug organ toxicity. Defects in the protein molecule might increase plasma concentrations of unbound drugs, causing an adverse response from the administered treatment or severe toxic events, even when total albumin concentrations are within the reference range. As a result of defects in the albumin molecule and hypoalbuminemia, higher clearance and inability of adequate active transport of drugs, general drug intoxication, longer hospitalization, ineffective therapy, and high mortality might be observed [10].

## 2. Conventional Colorimetric Serum Albumin Level Tests vs. SDSL-EPR Spectroscopy

### 2.1. UV-Visible Spectroscopy and Colorimetric Tests—Limitation

Currently, there are various methods for determining serum albumin levels that are part of routine clinical and biochemical laboratory analyses, which are based on measuring the absorbance of the protein-dye complex at a certain wavelength. By their nature, they are spectrophotometric analyzes and involve the use of specific dyes (bromocresol purple (BCP), bromocresol green (BCG), etc.).

Although widely used in clinical laboratories, they have some limitations and drawbacks. For example, conventional colorimetric assays require maintaining a specific pH range, necessitating the use of additional reagents to maintain it. For example, in order to form a complex between BCG and albumin, it is necessary to maintain a certain pH (pH 4.2), which could potentially avoid the risk of false-negative or false-positive results [11]. BCG and BCP are characterized by a different analytical reading range and a large coefficient of variation. Differences in results between the BCG and BCP methods preclude the use of common reference intervals for interpreting results [12].

The indicator dyes used are not highly specific in terms of binding to the protein. This is a known drawback and may lead to an inaccurate analysis due to insufficient specificity of the dye to the protein. The standard spectrophotometric BCG method is affected by positive interference from α-1 and α-2 globulins, which contributes to the overestimation of albumin concentration in the acute phase of some diseases. The BCP method is more specific than BCG but underestimates serum albumin concentrations in patients with renal failure and those undergoing hemodialysis and peritoneal dialysis, and often the interfering substance is not clearly identified [13].

Spectrophotometric methods are sensitive to analytical variations in protein and albumin measurement affects the calculation of calcium levels (Total calcium (TCa)/ionized calcium (iCa^2+^). With a calcium concentration outside the reference range, the possibility of analytical variations necessitates the use of locally obtained equations, which must be specifically developed for the respective laboratory, and analysis of results based on the literature data is inapplicable [14]. Moreover, in some diseases such as hypertension, diabetic nephropathy, and liver cirrhosis, these variations create the premise for inaccurate measurement of small deviations in serum albumin, which are particularly decisive in the pathophysiological evaluation of disease states and can delay the start of treatment or worsen the general condition of the patient [13].

Standard colorimetric tests can only measure serum albumin levels but are unable to detect disorders in the structural flexibility of the protein molecule.

### 2.2. SDSL–EPR in Impaired Structural Flexibility of Albumin and Hypoalbuminemia Cases—The Potential and Perspective

SDSL–EPR is a universal technique that is used as an important tool for understanding the structure of various macromolecules, which allows to detection of subtle changes in the local conformational dynamics of proteins under different conditions [15,16,17,18,19,20]. It involves the covalent attachment of a paramagnetic label (called a spin-label, SDSL) to the protein molecule, which selectively reacts with the thiol groups in the cysteine residue, forming covalent S-single bond-S or C-single bond-S bonds [15]. In order to carry out a reaction between the protein and the SDSL, the spin label must have a stable paramagnetic moiety and include a specific functional group capable of selectively reacting with the amino acids in the protein molecule without disturbing its structure [21]. The most commonly used specific spin labels that meet these conditions are the stable nitroxide radicals, 3-maleimido-2,2,5,5-tetramethyl-1-pyrrolidinyloxy (3-maleimido-PROXYL, 5-MSL), 2,2,5,5-tetramethyl-1-oxyl-3-methyl methanethiosulfonate (MTSL), bifunctional derivatives [22]. They are stable paramagnetic radicals that are not affected by perturbations in biological systems such as polarity, changes in the size and conformation of biomolecules (proteins, nucleic acids, lipids, and enzyme substrates), spatial restrictions, and the presence of another paramagnetic center [23].

Since nitroxide radicals are capable of specific and selective covalent binding to albumin molecules, this classifies them as suitable analytical sensors for measuring concentration and studying albumin protein structure and dynamics. SDSL—EPR excludes all the listed disadvantages of conventional methods and is characterized as a fast, specific, and highly sensitive method for determining the concentration and structural-dynamic changes in macromolecules [22,23]. Unlike the conventional spectroscopic methods used up to now in clinical practice, EPR spectroscopy allows simultaneous proof of subtle conformational changes in the albumin molecule and altered structural flexibility (Figure 1) and at the same time allows determination of its serum concentration (Figure 2). Due to the high stability under physiological conditions, SDSLs are rapidly becoming valuable biochemical tools in the measurement of the dynamic mobility of molecules such as phospholipids, peptides, proteins, and drugs in biological systems, including in membranes and whole protein sequence mapping [22,24].

## 3. Application of Stable Nitroxide Radical 3-Maleimido-PROXYL in the Preclinical Diagnosis of Albumin Concentration and Flexibility in Patients with COVID-19 in Critical Condition

In the study of protein structure and determination of conformational changes in macromolecules (proteins, DNA, RNA, and lipids) are applied various EPR approaches such as CW-EPR (continuous-wave electron paramagnetic resonance spectroscopy), DEER-EPR (double electron-electron resonance), and DQ-EPR (double quantum EPR) [15].

### 3.1. Identification of Albumin Structural Flexibility by Spin Labeling EPR Spectroscopy

The attachment of a paramagnetic label (3-Maleimido-PROXYL; 16-DOXYL-stearic acid, etc.) on a specific position in the protein is due to a selective reaction of the nitroxide with the thiol group of the cysteine residue in the side chains of proteins [20]. The achieved covalent binding allows the assessment of albumin conformational changes in critically ill patients with COVID-19 and shows excellent agreement between the parameters of the EPR spectrum of the radical and perturbations in the albumin structure. The diagnostic utility of the EPR method is performed by tracking the spectral parameters A_max_ and ΔH_0_ in a 3-Maleimido-PROXYL/DMSO control sample (A), healthy volunteers (B), and patients with COVID-19 (C) (Figure 1).

The EPR parameters A_max_ and ΔH_0_ of 3-Maleimido-PROXYL/DMSO showed the free rotation of the radical and is represented by the typical nitroxide triplet (A). In the cases of clinically healthy volunteers, albumin concentrations above 3.5 g/dL (protein levels in the reference range of 3.5–5.2 g/dL) and without protein structural changes, lead to immobilization of spin-marker rotation observed due to the nitroxide attachment to the albumin (B). The EPR parameters Amax and ΔH_0_ measured in patients with COVID-19 are close in values with 3-Maleimido-PROXYL/DMSO, and the spectrum of PROXYL radical is identical and comparable to that of the radical in DMSO. In the COVID-19 cases (C), probably due to low serum albumin levels (<3.5 g/dL, hypoalbuminemia), changes in protein conformation and dynamics imply the free rotation of 3-Maleimido-PROXYL, comparable to that of nitroxide in DMSO (protein-free sample).

The inflammation caused during the cytokine storm alters the redox balance in the body, leading to epithelial, endothelial, mitochondrial, metabolic, and immune dysfunction. Low levels of nitric oxide, compromised redox regulation, and endogenous antioxidant defenses render the sulfur-based redox chain vulnerable to oxidation [9,25]. High levels of ROS lead to impaired structural flexibility of albumin in critically ill patients with COVID-19. This limits the binding of nitroxide radical 3-maleimido-PROXYL to Cys^34^ in the protein molecule (Figure 1C).

### 3.2. Determination of the Serum Albumin Concentration by Site-Directed Spin Labeling EPR

Serum albumin levels are used as a predictor in determining the severity of acute viral infections and have a high significance in relation to the survival of critically ill patients. Hypoalbuminemia is observed in more than 70% of critical cases of SARS-CoV-2 disease, which is a marker of low survival [26]. The pharmacokinetics and toxicity of many compounds are influenced by the serum concentrations of this critical protein, with hypoalbuminemia patients having more prolonged clearance to xenobiotics and adverse drug reactions [27]. The spin-labeling EPR method can be applied as a direct non-invasive approach for monitoring HAS levels in both healthy humans and critical COVID-19 patients. It could be used for the early detection of hypoalbuminemia in various congenital or pathological conditions to prevent low albumin-related intoxication during standard drug therapy.

The double integrated area (DIA) of the EPR spectrum (Figure 2) expresses the protein concentration and depends on the molar ratio of 3-Maleimido-PROXYL and albumin. Protein—PROXYL interaction in healthy controls (3.5–5.2 g/dL, HAS) resulted in lower DIA values due to loss of nitroxide radical paramagnetism. Low albumin levels in critically ill patients with COVID-19 (HAS < 3.5 g/dL) correspond to higher DIA values. Retention of spin label paramagnetism was recorded and results were close to the 3-Maleimido-PROXYL/DMSO control (HAS = 0.0 g/dL).

The different sources of OS in COVID-19 include the generation of free radicals as a protective response to the entry of the virus into the body, an inflammatory response and a cytokine storm accompanied by the production of oxidants leading to the disruption of cellular structures [1,25] and irreversible protein oxidation [8]. The therapeutics used in the treatment of COVID-19 can exacerbate the pathogenesis of COVID-19, directly increase the risk of intoxication [28], and disrupt the normal redox balance in the body by mediating the formation of ROS/RNS [29].

As a rule, the establishment of hypoalbuminemia requires intravenous administration of human serum albumin in critically ill COVID-19 patients [30]. This may be insufficient and ineffective because, as a result of OS, circulating plasma protein has impaired protein dynamics and should be completely replaced by serum albumin without structural and conformational changes. In order to prevent the body’s unfavorable response, there is a need for an unerring measurement of serum albumin levels, but at the same time an assessment of albumin structural flexibility and dynamics.

It is necessary that the methods for measuring HAS are highly sensitive because small deviations in the values can compromise the implementation of timely and adequate therapy, especially one involving drugs with high protein binding. This creates conditions for the low effectiveness of drug therapies and increased drug toxicity, which in turn necessitates an urgent correction of the used doses.

## 4. SDSL-EPR Spectroscopy Limitation

Chemoselectivity is a key factor in the application of SDSL-EPR spectroscopy in complex biological environments, especially in the context of target protein in vitro. Low conjugation rates or low concentrations of reaction products can compromise the efficient labeling of proteins. The tags used and the stability of the linker formed can be conformationally incompatible, and excessively long and/or overly flexible linkers can create the premise for inaccurate information [31]. In order to exclude minimal structure-function perturbations and increase the sensitivity and performance of SDSL-EPR analysis, it is crucial to optimize the process of introducing the nitroxide spin labels during sample preparation [23], to choose an appropriate experimental protocol by determining the most suitable conditions for conducting the EPR study [14].

### 4.1. Type of Spin Marker

The nitroxide spin labels are kinetically or sterically stabilized by carbon substituents in the α-position relative to the nitrogen atom, which hinders the reduction of the nitroxide. The stable structure provides conformational flexibility, but the presence of large steric substituents in the radical increases the potential of bulky spin labels to initiate structural and functional perturbations in the labeled protein [23]. Typically, the spin-labeling reaction is slow and requires a large excess of reagents that must subsequently be removed [32]. The excess unreacted label is removed by scanning each fraction of the sample until nitroxide is completely eliminated. The signal from the free label can be difficult to read accurately, especially in the case of small, highly dynamic peptides, which highlights the need for careful separation of the free label from the labeled protein [33]. Hideg et al. developed specific alkylthiosulfonate spin labels, which are distinguished by their extremely fast reactivity under mild conditions, and high selectivity for sulfhydryl groups, without requiring a large excess of the range [34].

When studying the membrane topology, double labeling techniques (DEER) are applied, which allows measuring the distances between two spin labels and determining intramolecular distances of the same protein as well as intermolecular distances between sites of different proteins. The application of DEER spectroscopy has some limitations. The non-uniform distribution of spin-labeled proteins and the occurrence of local inhomogeneous pockets of high spin concentration create conditions for much shorter relaxation times and poor modulation of DEER, e.g., in liposomes, compared to water-soluble proteins. A highly effective protein concentration introduces a strong background contribution that can seriously reduce the sensitivity and performance of DEER [35]. The nitroxide labels used in DEER have a fast relaxation rate at room temperature, so measurements are usually performed at cryogenic temperature. Another limiting factor is that they possess a phase memory on the order of Tm < 2 µs, fixing the maximum of detectable interspin distances at around 6–8 nm [14]. Different groups are striving to reduce the limitations of SDSL-EPR in studying complex biological systems by modulating the protein/lipid molar ratio, using nanoparticles as carriers, synthesizing new nitroxide spin-labels, developing simulation programs, etc., to increase the applicability of EPR methods [35,36,37].

### 4.2. The Concentration of the Studied Protein

The very low concentrations of the observed proteins in cells represent a major limitation for standard SDSL-EPR. This necessitates the use of expressed protein ligation and allows the semi-synthesis of proteins from recombinant and synthetic fragments. The approach aims to expand the size and complexity of protein targets and involves custom-synthesized spin-labeled Rasbinding domains (RBDs) with a stable paramagnetic center that can be detected by EPR [16].

### 4.3. Formation of Protein Substrates

Proteins are inherently dynamic structures that often exhibit a number of conformational substates involved in their function, such as proteins with a dynamic portal region with open and closed states of a ligand binding pocket. The half-lives of these states range from μs to ms. Observing slow (μs) movements requires the use of rigid nitroxide side chains [17]. The formed substrates can be detected using conventional EPR spectroscopy at room temperature. Multicomponent CW-spectra are obtained, which are due to the different mobility of the side chain of the spin label. These are due to the different linker lengths and conformational flexibility of the spin-label side chains and arise as a result of protein conformational substates, which will lead to different rotameric states of the spin-label. To distinguish the protein substrates from the rotameric states of the spin labels, additional techniques such as NMR spectroscopy and high-pressure EPR are required [16].

### 4.4. Limited Accessibility of Cys-SH

A significant limitation of conventional SDSL can occur when the cysteine residue in the protein is involved in the function and/or structural elements (active sites, disulfide bridges). If the cysteine residue is involved in disulfide bridges or is coordinatively bound to a metal cofactor, spin-label binding cannot occur at Cys-SH. In such cases, a nitroxide that targets a non-cysteine residue can be used [36]. In stable well-folded proteins, EPR spectra cannot provide detailed direct information about the backbone dynamics of the protein molecule because the amplitudes are small and the parameters are limited to a narrower range. Backbone motions are not transmitted directly to the nitroxide ring due to the damping effect of the intervening flexible tag side chain. Therefore, CW-SDSL-ESR is not particularly applicable in quantifying small variations in protein backbone parameters within a rigid protein scaffold. Bifunctional spin tags can increase the sensitivity to backbone dynamics but may cause perturbations in the natural dynamics of the system [38]. The low redox stability of disulfide bonds in the cell represents another obstacle for SDSL. This can be corrected by peptide ligation or the application of chelating protein tags, but this may limit the choice of labeling site and complicate technical implementation [32].

### 4.5. Protein and Nitroxide Stability—Solvents, Temperature, pH

Sample processing for EPR spectroscopy is a key factor in the study of protein dynamics and should be considered when choosing an EPR methodology. Protein labeling involves the pre-preparation of a spin-label stock solution in a solvent (acetonitrile, ethanol, or DMSO). In order not to disturb the stability of the nitroxide, the stock solution should be stored at or below −20 °C and protected from light. The size of the protein is important, as well as the removal of the free label since an overlap of the signals from the labeled protein in the solution and the unremoved nitroxide label can be observed [33]. The presence of water molecules can lead to the non-resonant microwave absorption and non-resonant absorption/heating of water that occurs in much of the microwave region, especially in cell studies. Various physicochemical processes or mechanical forces involved in the sample preparation process can produce free radicals [18] or be a source of artifacts to be interpreted as part of dipolar oscillations. This can lead to the extraction of erroneous distance distributions and misinterpretation of the results obtained [39]. In some cases, the use of reducing agents dithiothreitol (DTT, D1532), 2-mercaptoethanol (b-mercaptoethanol), or tris [2-carboxyethyl] phosphine (TCEP) is required. The purpose of such reducing agents is to prepare the thiol group for selective tag attachment. After the reduction process, reducing agents must also be removed as they may compete with target thiols in proteins, although their removal may be accompanied by oxidation of the thiols back to disulfide [40].

SDSL-EPR is often performed at room temperature [15], or at lower (e.g., 4 °C, including cryogenic temperatures. Here a limitation may occur depending on the stability of the protein at different temperatures [20]. Maleimide spin labels react with thiols at room temperature or below and at pH in the range of 6.5–8.0. If labeling is performed at elevated pH, thiol-specific spin labels can react with amines or hydrolyze to unreactive products, which compromise labeling specificity [32]. Pavićević et al., labeled bovine serum albumin (BSA) with 3-Meleimido-PROXIL and monitored the EPR parameter Amax at pH in the range of 2–11. Depending on the pH, BSA undergoes unfolding and loosening of its structure. In the pH range of 4.5–8.0, the hydrophobic cleft of Cys-34 is very stiff, resulting in Amax reaching a plateau. Outside this range Amax decreases, suggesting that the gap is opened and becomes more flexible as a result of acid–base transitions. The nitroxide group of 5-MSL is located deep in a hydrophobic cleft of Cys-34, so the radical is not sensitive enough to reflect these acid-base transitions. However, 5-MSL may be a suitable spin-marker under pH-controlled conditions and should not be excluded when investigating conformational changes in serum albumin [19].

The listed limitations can be overcome to one degree or another, by choosing appropriate spin labels and methodological protocols that are tailored to the type of medium (membranes, aqueous solution, etc.) [33,41], the type of protein studied (globular, membrane and intrinsically disordered proteins) [15,23], reaction time, chain mobility and nitroxide side chain dynamics [42], pH and temperature [17,43], the parameters at which the scan will be performed, etc. [44].

## 5. Conclusions

ROS modifies the amino acid residues of the vital transport protein—albumin and initiates structural and functional disorders in its molecule, which can lead to general drug intoxication. Site-Directed Spin Labeling EPR is suitable for investigating of various protein processes and interactions, cofactor effects, and protein concentration monitoring. The present opinion aims to demonstrate the diagnostic utility and application of the SDSL-EPR method in the study of hypoalbuminemia and changes in albumin dynamics in cases of critically ill patients with COVID-19. Despite existing limitations, EPR spectroscopy can solve some biologically important problems and reveal important structural-dynamic information about protein systems in solution or membrane-bound states, which is beyond the scope of conventional biophysical and spectrophotometric techniques.

## 6. Patents

The patents resulting from the work reported in this manuscript is with the incoming number of the patent application: BG/U/2022/5487.

## Figures and Tables

**Figure 1 antioxidants-11-02311-f001:**
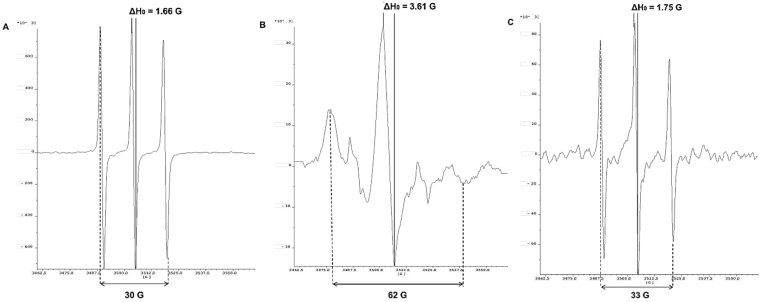
3-Maleimido-PROXYL radical spectra: Control of 3-Maleimido-PROXYL/DMSO without serum (**A**); in serum samples of healthy patients (**B**); in serum samples in patients with COVID-19 (**C**).

**Figure 2 antioxidants-11-02311-f002:**
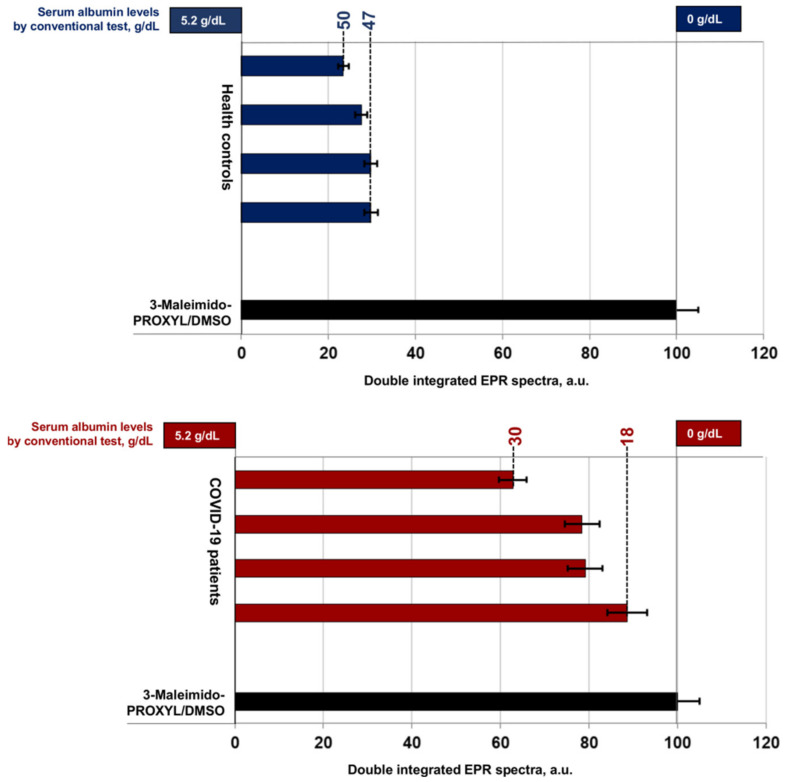
Double integrated EPR spectra of 3-Maleimido-PROXYL radical presented as percent value of control: 3-Maleimido-PROXYL/DMSO without serum (in black); healthy volunteers (in blue); in patients with COVID-19 (in red).

## Data Availability

The personal data presented in this study are not publicly available due to privacy restrictions. The data are available on request from the corresponding author.

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
