# Peer review of "Site-Directed Spin Labeling EPR Spectroscopy for Determination of Albumin Structural Damage and Hypoalbuminemia in Critical COVID-19"

_antioxidants, 2022, doi:10.3390/antiox11122311_

Round 1
Reviewer 1 Report
This opinion article demonstrates that extensive structural changes at the cellular and molecular levels such as the exacerbated formation of free radicals (ROS) can generate important factors in the pathogenesis of COVID-19. These mechanisms can induce changes in different proteins that will drastically affect cellular functionality, gradually altering homeostasis.
Among all the possible proteins in the human body, this opinion article demonstrates the role of serum albumin, a protein that has a very high binding capacity and has an optimal concentration for different factors related to the functioning of the organism. The importance of this molecule can be seen with the complications related to changes in normal concentrations.
The authors argue that structural albumin and functional changes accompanied by low protein levels may be a serious prerequisite for ineffective therapy, frequent complications, and high mortality in patients with SARS-CoV-2 infection.
The opinion article provides a pleasant and important discussion for the field of study.
Author Response
Dear reviewer,
Thank you very much for your opinion.
Reviewer 2 Report
The authors propose to use more specific site-directed spin labeling electron paramagnetic resonance (EPR) to characterize albumin structural changes in COVID-19.
While the authors stated that this approach improved sensitivity and might have clinical implications, additional experiments are needed to support their conclusion.
1. Hypoalbuminemia is a non-specific phenomenon in critically ill patients, not limited to COVID-19 patients. It would be helpful to measure albumin using this approach for non-COVID infection or non-COVID critical ill cases. The differences between COVID and non-COVID cases would improve specificity.
2. Albumin structural changes seemed to occur preceding a reduction of albumin level. Longitudinal measurement in different clinical stages would also be helpful.
3. Albumin transfusion has never been shown to improve outcomes in critically ill patients except in a setting of liver cirrhosis. It also has never been shown to worsen mortality. The notion that albumin infusion could result in adverse clinical outcomes due to OS state that can increase ROS needs a reference to support.
Author Response
Dear reviewer,
Thank you very much for helping us to improve our manuscript Antioxidant 1986575, titled: “Site-Directed Spin Labeling EPR Spectroscopy for Determination of Albumin Structural Damage and Hypoalbuminemia in Critical COVID-19”
Please find attached our answer.

Reviewer 3 Report
The precise measurement the concentration and structural integrity of albumin is very important not only in the scientific research but also in the clinical diagnosis and treatment. The reviewer think that this manuscript would be better if it is re-written for 'research article' or 'short communication'.
Author Response
Dear reviewer thank you for your constructive review and your time, we appreciate your experience and expertise in the field.
Point 1: “The precise measurement of the concentration and structural integrity of albumin is very important not only in the scientific research but also in the clinical diagnosis and treatment. The reviewer think that this manuscript would be better if it is re-written for a 'research article' or 'short communication'.”
Answer 1: The problem we have indicated and the possibility of solving it is structured in the form of "Opinion" because the opinion presented by us meets the requirements of MDPI for Article type "Opinion". It represents short articles with a review structure that reflect our viewpoints on technique or recent findings. We tried to present the strengths and weaknesses of conventional spectrophotometric tests for albumin analysis and a possible new approach involving electron paramagnetic resonance spectroscopy (SDSL-EPR) use.
The Opinion's purpose is to urgently draw the attention of the scientific community to the problem under our consideration, namely that low levels of albumin and/or impaired protein structure may lead to ineffective therapy, further worsening the already critical condition of patients with COVID-19. At the same time, we want to emphasize the relevance and analytical possibilities of spectroscopic methods and stable nitroxide radicals, as useful tools in cases of limitations in conventional methods (for example, in the determination of structural instability of protein molecules - albumin). We present an opinion that the SDSL - EPR method can allow early detection of impaired protein binding capacity (protein dysfunctionality, including subtle changes caused by protein-ligand binding). In our opinion manuscript, the SDSL-EPR method may be a useful tool in cases of pre-critical and/or critical COVID-19 patients with hypoalbuminemia or those with albumin levels still within the reference range (3.5-5 .2 g/dL). At the same time, and based on our ongoing research, we hypothesize that SDSL-EPR spectroscopy will allow not only the determination of a disordered structure of the protein molecule but also the accurate determination of serum albumin concentrations.
Currently, we are in the process of developing a new methodology for the analysis of hypoalbuminemia in patients with COVID-19, using Electron Paramagnetic Resonance Spectroscopy and the stable spin label 3-Melaimido-PROXYL. At present, we are continuing our work in this area by simultaneously conducting toxic-chemical analysis of blood and urine samples of the patients included in the study. The inclusion of different approaches and methods implies a long and responsible process of full-scale research, and soon, according to your guidelines and recommendations, we will provide the scientific community with the results of our research, which will be described in scientific articles.
Reviewer 4 Report
In this opinion, the authors recommended using sire-directed spin labeling EPR spectroscopy with 3-Maleimido-PROXYL radical for evaluating the structure damage and concentration of albumin in critical COVID-19 patients. Compared with conventional colorimetric test and spectrophotometric method, SDSL-EPR has more advantages, making it more suitable for hypoalbuminemia cases with impaired structural flexibility. My suggestions for revision are as follows.
1. Line 145-151, the topic title is “Application of stable nitroxide radical 3-Maleimido-PROXYL in the preclinical diagnosis of albumin concentration and flexibility in patients with COVID-19 in critical condition.” However, there is no related information in this paragraph. And the Reference [18] has nothing to do with application of Maleimido-PROXYL in patients with COVID-19. So, this paragraph should be rewritten to supplement relevant information.
2. I am wondering whether SDSL-EPR has any limitations? SDSL-EPR has more advantages than conventional method from the authors’ point of view. Each method has inherent limitation. The readers would get more comprehensive understanding about SDSL-EPR if its limitations are discussed.
3. The full name of EPR should be provided when it first appears, even if we experts know what it is.
4. Line 59, something wrong with this sentence?
5. Line 116-118, some literatures with great significance should be cited to display the application of SDSL-EPR in understanding the structure of various macromolecules.
Author Response
Dear reviewer,
Thank you for help us to improve our manuscript.
All corrections according to your remarks are in red in a manuscript.
Point 1: Line 145-151, the topic title is “Application of stable nitroxide radical 3-Maleimido-PROXYL in the preclinical diagnosis of albumin concentration and flexibility in patients with COVID-19 in critical condition.” However, there is no related information in this paragraph. And the Reference [18] has nothing to do with application of Maleimido-PROXYL in patients with COVID-19. So, this paragraph should be rewritten to supplement relevant information.
Answer: Done
Point 2: I am wondering whether SDSL-EPR has any limitations? SDSL-EPR has more advantages than conventional method from the authors’ point of view. Each method has inherent limitation. The readers would get more comprehensive understanding about SDSL-EPR if its limitations are discussed.
Answer: Done
All cited literature is added in the manuscript.
Point 3: The full name of EPR should be provided when it first appears, even if we experts know what it is.
Ansewr: Done
Point 4: Line 59, something wrong with this sentence?
Author response: Done
Point 5: Line 116-118, some literatures with great significance should be cited to display the application of SDSL-EPR in understanding the structure of various macromolecules.
Answer 5: Done
Round 2
Reviewer 2 Report
The authors have extensively responded to the comments. It was greatly appreciated.
1. It was stated that their study included some critically ill non-COVID-19 patients. If possible, please report the number of patients and please include the data of these patients in Figures 1 and 2. Currently, those figures only compare healthy controls and critically ill COVID-19 patients.
2. In the conclusion section, given that there are many limitations of this method SDSL-EPR and a limitation of clinical data to support its sensitivity and specificity for COVID-19 patients, the authors should modify their conclusion.
Author Response
Point 1: It was stated that their study included some critically ill non-COVID-19 patients. If possible, please report the number of patients and please include the data of these patients in Figures 1 and 2. Currently, those figures only compare healthy controls and critically ill COVID-19 patients.
Answer 1: Only patients with COVID-19 are included in the presented opinion. All other critical conditions characterized by hypoalbuminemia outside of cases of COVID-19 (non-COVID-19) are not the focus of this Opinion.
Point 2. In the conclusion section, given that there are many limitations of this method SDSL-EPR and a limitation of clinical data to support its sensitivity and specificity for COVID-19 patients, the authors should modify their conclusion.
Answer 2: Done (included in the manuscript in red)
Reviewer 3 Report
Since the manuscript has been improved a lot and the significance of the measurement of albumin concentration and the its novel methodology were described well, the manuscript is publishable for the 'opinion' in Antioxidants.
Author Response
Dear Reviewer,
Thank you very much for helping us make our manuscript better.
Reviewer 4 Report
The manuscript improved a lot after revision.
Author Response

(The authors gave the same response as above.)
